# HDAC Inhibitory and Anti-Cancer Activities of Curcumin and Curcumin Derivative CU17 against Human Lung Cancer A549 Cells

**DOI:** 10.3390/molecules27134014

**Published:** 2022-06-22

**Authors:** Narissara Namwan, Gulsiri Senawong, Chanokbhorn Phaosiri, Pakit Kumboonma, La-or Somsakeesit, Arunta Samankul, Chadaporn Leerat, Thanaset Senawong

**Affiliations:** 1Department of Biochemistry, Faculty of Science, Khon Kaen University, Khon Kaen 40002, Thailand; narissaranamwan@kkumail.com (N.N.); gulsiri@kku.ac.th (G.S.); s_arunta@kkumail.com (A.S.); l_chadapornee@kkumail.com (C.L.); 2Department of Chemistry, Faculty of Science, Khon Kaen University, Khon Kaen 40002, Thailand; chapha@kku.ac.th; 3Department of Applied Chemistry, Faculty of Science and Liberal Arts, Rajamangala University of Technology Isan, Nakhon Ratchasima 30000, Thailand; pakit.ku@rmuti.ac.th; 4Department of Chemistry, Faculty of Engineering, Rajamangala University of Technology Isan, Khon Kaen 40000, Thailand; laor.so@rmuti.ac.th

**Keywords:** apoptosis, cell cycle arrest, curcumin derivative, HDAC inhibitor, lung cancer, molecular docking

## Abstract

Previous research reported that the curcumin derivative (CU17) inhibited several cancer cell growths in vitro. However, its anticancer potential against human lung cancer cells (A549 cell lines) has not yet been evaluated. The purpose of this research was to examine the HDAC inhibitory and anti-cancer activities of CU17 compared to curcumin (CU) in A549 cells. An in vitro study showed that CU17 had greater HDAC inhibitory activity than CU. CU17 inhibited HDAC activity in a dose dependent manner with the half-maximal inhibitory concentration (IC_50_) value of 0.30 ± 0.086 µg/mL against HDAC enzymes from HeLa nuclear extract. In addition, CU17 could bind at the active pockets of both human class I HDACs (HDAC1, 2, 3, and 8) and class II HDACs (HDAC4, 6, and 7) demonstrated by molecular docking studies, and caused hyperacetylation of histone H3 (Ac-H3) in A549 cells shown by Western blot analysis. MTT assay indicated that both CU and CU17 suppressed A549 cell growth in a dose- and time-dependent manner. Besides, CU and CU17 induced G2/M phase cell cycle arrest and p53-independent apoptosis in A549 cells. Both CU and CU17 down-regulated the expression of p53, p21, Bcl-2, and pERK1/2, but up-regulated Bax expression in this cell line. Although CU17 inhibited the growth of lung cancer cells less effectively than CU, it showed less toxicity than CU for non-cancer cells. Accordingly, CU17 is a promising agent for lung cancer treatment. Additionally, CU17 synergized the antiproliferative activity of Gem in A549 cells, indicating the possibility of employing CU17 as an adjuvant treatment to enhance the chemotherapeutic effect of Gem in lung cancer.

## 1. Introduction

Lung cancer (LC) is one of the most common cancer incidences worldwide and in Thailand [1]. It is also the cause of the highest number of cancer-related deaths (21%) in both males and females in the United States [1]. Non-small cell lung cancer (NSCLC) is the most common type of lung cancer, accounting for 80–85% of all cases [2]. This is followed by small cell lung cancer (SCLC), which accounts for 15–20% of all cases [2]. There are several other factors that can cause lung cancer. However, tobacco smoking is the leading major cause of lung cancer [3]. Chemotherapeutic regimens have recently been the main therapeutic strategy for lung cancer. The most effective chemotherapeutic drugs for NSCLC include cisplatin, docetaxel, gemcitabine (Gem), irinotecan, and paclitaxel [2]. Nevertheless, chemotherapeutic agents are nonspecific in their mechanism of action. They target key components or metabolic pathways of both malignant and normal cells, leading to many negative side effects [2]. Hence, the development of novel molecular mechanisms for lung cancer treatment is a major challenge, to improve agents with a greater cytotoxic effect on cancer cells but being safer for normal cells.

Histone acetylation is related to transcriptional activation and is stimulated by histone acetyl transferases (HATs), whereas histone deacetylation is related to transcriptional repression and is activated by histone deacetylases (HDACs) [4,5]. Imbalances in these enzymes may cause increased proliferation and differentiation in normal cells, which can contribute to the development and progression of cancer. HDAC dysregulation is essential for most of the formation and progression of various human cancer types [6]. 

HDAC inhibition reduced the proliferation of malignant cell lines through the triggering of cell cycle arrest and apoptosis [7]. HDAC inhibitors (HDACis) exhibit potent anticancer effects on a wide range of cancer cells by causing apoptosis, differentiation, and inhibition of tumor angiogenesis [8]. Currently, there are four varieties of HDACis, including vorinostat, romidepsin, belinostat, and panobinostat, which have been authorized by the US Food and Drug Administration [9]. However, during the treatment of solid tumors in clinical trials, high doses of these HDACis led to some negative side effects occurring in LC patients [10]. Therefore, numerous research interests are focused on creating novel HDACis for cancer therapeutic strategies [11]. Over the last decade, many isoform- or class-selective HDAC inhibitors with various capping groups have been synthesized, providing a promising way to overcome drug toxicity and side-effect issues [12].

Curcumin (CU) is a nutraceutical product extracted from rhizomes of *Curcuma longa* in the ginger family. It is a polyphenol compound that has several beneficial biological activities including antioxidant, anti-inflammatory, anti-proliferative, and antiangiogenic properties. Furthermore, CU is identified as a member of HDAC inhibitors used as an anti-cancer agent inducing Raji cell apoptosis [13]. A half maximal HDAC inhibitory concentration was reported at a level of micromolar (IC_50_: 115 µM) [14]. Previous reports showed that CU induced cytotoxicity in NSCLCs via the inhibition of cell proliferation and differentiation, induction of apoptosis, induction of cell cycle arrest at G2/M phases, and induction of ROS production, contributing to the activation of the DNA damage signaling pathway [15]. Nevertheless, the clinical applications of CU are limited by its poor water solubility, rapid metabolism, and high toxicity in non-cancer cells. Accordingly, much attention has been directed towards searching for new chemically synthetic curcumin analogues with similar biological activities. Recently, there were several reports on the anti-proliferation research on curcumin analogues or derivatives in various cancer cells [16,17,18,19,20]. For instance, HHMM-41 and JZ534 curcumin derivatives exhibited higher inhibition of cell proliferation than curcumin in lung cancer cells with no significant cytotoxicity in normal cells [16,17].

Recently, several amino derivatives of curcumin (CU16, CU17, CU18 and CU19) with lower physiochemical instability but stronger pharmacological activities than curcumin have been synthesized [21]. In this present research, the curcumin derivative CU17 with 2-aminothiophenol (Figure 1) was chosen for further studies on its HDAC inhibitory and anti-cancer activities against the NSCLC cell line because it presented the least toxicity to non-cancer cells (Vero cells) and great efficiency of anti-proliferative activity against several cancer cell lines [21]. At the screening concentration of 100 µM, CU17 had greater HDAC inhibition (73%) than curcumin (62%) [21]. However, its anti-cancer activity against lung cancer cells has not yet been investigated. Therefore, the aims of this study were to further investigate HDAC inhibitory activity (in vitro and in silico) and the anti-cancer activity of CU17, as well as exploring its underlying anti-cancer mechanism against human lung cancer A549 cells.

## 2. Results

### 2.1. HDAC Inhibitory Activity of the Curcumin Derivative CU17

HDAC inhibitory activity of CU17 was measured in vitro by using the Fluor-de-Lys HDAC Fluorometric Activity Assay Kit. As shown in Figure 2, CU17 inhibited HDAC activity in a dose dependent manner with the half-maximal inhibitory concentration (IC_50_) value of 0.30 ± 0.086 µg/mL (0.41 ± 0.10 µM) against human HDAC enzymes. In addition, in silico molecular docking studies were also performed to support the in vitro results and to understand the inhibitory activity of CU17 against HDAC isoenzymes. Molecular docking was performed, employing CU17 as a ligand, docking with the crystal structures of human class I HDACs (HDAC1, 2, 3, and 8) and class II HDACs (HDAC4, 6, and 7). As shown in Table 1, the binding energy (ΔG) and inhibitory constant (K*_i_*) of CU17 indicated that CU17 enhanced HDAC inhibitory activity against class I HDACs, including HDAC1 (∆G = −8.91 kcal/mol, K*_i_* = 0.29 µM), HDAC2 (∆G = −7.64 kcal/mol, K*_i_* = 2.51 µM), HDAC3 (∆G = −8.19 kcal/mol, K*_i_* = 0.99 µM), and HDAC8 (∆G = −7.96 kcal/mol, K*_i_* = 1.45 µM). Furthermore, CU17 revealed HDAC inhibitory activity more effectively on class II HDACs, including HDAC4 (∆G = −9.15 kcal/mol, K*_i_* = 0.20 µM), HDAC6 (∆G = −9.21 kcal/mol, K*_i_* = 0.18 µM), and HDAC7 (∆G = −8.40 kcal/mol, K*_i_* = 0.70 µM). As shown in Figure 3, in silico data demonstrated that CU17 is proposed to interact with several amino acids in the active sites of the enzymes, while in silico data of CU were provided in the Appendix A. CU17-HDAC1 complex exhibited a zinc ion coordination (3.67 Å) and three hydrogen bonds occurred between those of CU17 with residues of Asp99 (2.13 Å, 2.77 Å, 1.96 Å), Gln26 (1.8 Å), and His28 (2.59 Å) in the active site of HDAC1. Additionally, CU17 interacts with Leu271 (2.06 Å, 2.52 Å) through the hydrophobic interaction. The HDAC2 binding site presented a zinc ion chelation (2.40 Å) and three hydrogen bonds and hydrophobic interaction occurring between CU17 and Gln265 (3.06 Å), Asp104 (2.12 Å), Arg275 (2.57 Å), and Gly143 (2.75 Å) residues, respectively. The binding interaction between CU17 and HDAC3 at active sites occurred via three hydrogen bonds and the hydrophobic formation of Asp93 (2.65 Å), Asp92 (2.10 Å), Asn197 (1.94 Å), and Phe200 (2.14 Å) residues, respectively. The binding mode of CU17 in the HDAC4 cavity occurred through two hydrogen bonds and two hydrophobic interactions of Asp90 (2.81 Å), His159 (2.51 Å), Pro298 (2.21 Å, 2.95 Å), and Leu299 (3.04 Å). In addition, CU17 coordinated with a zinc ion (2.4 Å) in the catalytic pocket of HDAC4. The major binding interaction between CU17 and HDAC6 is three hydrogen bonds via residues of Asp328 (2.61 Å, 2.92 Å), Ser150 (1.81 Å), and Trp261 (3.96 Å) at the active pocket of the enzyme. The CU17-HDAC7 complex revealed a zinc ion chelation (3.2 Å) and the hydrophilic and hydrophobic formation via Asn736 (2.70 Å), Pro739 (3.09 Å), and Pro809 (1.97 Å). CU17 binds to Tyr100 (1.85 Å) and Asp101 (1.90 Å) residues at the active pocket of HDAC8 via the formation of hydrogen bonds. Furthermore, hydrophobic interaction occurred between the CU17 and Ile94 (2.44 Å) and Leu98 (2.00 Å) residues at the HDAC8 active cavity. In addition, coordination between CU17 and the zinc ion occurred in all active pockets of HDACs, in which the zinc ion acts as a cofactor of HDACs. These results suggest that CU17 acts as an HDAC inhibitor, regulating the HDAC enzyme activity. 

An in vitro study of HDAC inhibition revealed that CU17 repressed HDAC activity in a dose-dependent manner (Figure 2a). Furthermore, we conducted Western blot analysis to investigate the HDAC inhibitory effect of CU17 in a human lung cancer cell line by measuring the acetylation condition of histone H3. As shown in Figure 2b,c, CU17 treatment triggered hyperacetylation of histone H3 in A549 cells. These results indicate that CU17 acts as an HDAC inhibitor as demonstrated by in silico docking and the in vitro study, supporting its activity in A549 cells.

### 2.2. The Anti-Proliferative Activities of CU17 against Human Lung Cancer A549 Cells

To investigate whether CU17 could promote the anti-cancer effect in human lung cancer cells (A549 cells) compared with curcumin (CU), A549 cells were treated with various concentrations of CU or CU17 (1.56, 3.13, 6.25, 12.50, 25, and 50 µg/mL) at 37 °C for 24, 48 and 72 h, and the anti-proliferative effects were evaluated by MTT assay.

As shown in Figure 4, CU and CU17 inhibited the proliferation of A549 cells in a dose and time dependent manner. The half-maximal inhibitory concentration (IC_50_) values of CU against A549 cells were 10.63 ± 0.74, 3.07 ± 0.15, and 2.24 ± 0.05 µg/mL, while the IC_50_ values of CU17 were 23.62 ± 0.54, 7.64 ± 0.43, and 4.64 ± 0.04 µg/mL, at exposure times of 24, 48 and 72 h, respectively. Nevertheless, CU treatment showed much stronger anti-proliferative effects than CU17 treatment.

The cytotoxic effects of CU and CU17 on non-cancer Vero cells were examined. The IC_50_ values of CU17 against Vero cells were >400, 18.66 and 11.11 µg/mL (Figure 4e), whereas the IC_50_ values of CU were 25.90, 11.29, 8.33 µg/mL, at exposure times of 24, 48 and 72 h, respectively. CU had a more cytotoxic effect on Vero cells than CU17 (Figure 4c), while the increased concentrations of CU17 did not inhibit the growth of Vero cells significantly at 24 h-exposure (Figure 4d). These results suggest that CU17 inhibits the proliferation of lung cancer cells less effectively than CU but exhibits less toxicity against non-cancer cells.

### 2.3. Effect of CU17 on the Cell Cycle Progression in Human Lung Cancer A549 Cells

Cell cycle arrest can cause the inhibition of cell proliferation. The effects of CU17 and CU on the cell cycle progression in lung cancer cells were comparatively analyzed by flow cytometry using PI staining. The dose of treatment was varied based on the IC_50_ values from the MTT assay to examine the cell cycle distribution in A549 cells. As shown in Figure 5, A549 cells exposed to CU for 24 h exhibited an increase in the percentage of cells halting at the G2/M phase, which was greater than that of the solvent control treatment. On the contrary, CU treatment resulted in a lower number of cells halting at the G0/G1 phase compared with that of the solvent control treatment. Additionally, CU treatments at 12.50 and 50 µg/mL enhanced the percentages of cells arresting at S phase. Moreover, incubation with 6.25 and 12.50 µg/mL of CU raised the populations of apoptotic cells in the subG1 phase (3.75% and 7.25%, respectively). 

CU17 treatment with a concentration of 50 µg/mL caused the decrease in cell percentage arresting at the G0/G1 phase (63.67%), but not with concentrations of 12.50 and 25 µg/mL (81.70% and 78.67%, respectively), when compared with the control treatment (78.15%). Furthermore, only 50 µg/mL CU17 treatment significantly increased the cell percentages arresting at S and G2/M phases (18.77% and 16.25%, respectively), and dramatically increased the populations of apoptotic cells in the subG1 phase (4.05%) compared with the control treatment (0.60%). These results indicated that both CU and CU17 promoted cell cycle arrest at S and G2/M phases in human lung cancer A549 cells. 

Based on the above cell cycle distribution results (Figure 5), cell cycle arrest at the G0/G1 phase was not observed, therefore, the expression of cell cycle-associated proteins was further determined. The p21 protein is a CDK inhibitor that prevents cell cycle progression by directly inhibiting the kinase activity of cyclin-CDK complexes (Cdk4/cyclin D and Cdk2/cyclin E) causing cell cycle arrest at the G0/G1 phase. Furthermore, the p53 protein regulated the amount of p21 expression. Hence, the expression of p21 and p53 proteins was determined by Western blot analysis. As shown in Figure 6, both CU and CU17 treatments resulted in a reduction in p21 and p53 protein expression compared with the solvent control treatment. In contrast, the expression levels of p21 and p53 were up-regulated in the treatment with cisplatin. In summary, no cell cycle arrest at the G0/G1 phase in the A549 cells is correlated with the expression levels of p21 and p53. 

### 2.4. Effect of CU17 on Induction of Cellular Apoptosis against Human Lung Cancer A549 Cells

The annexin-V/PI double staining assay was used to examine cellular apoptosis induced by CU and CU17 in A549 cells. As shown in Figure 7a,b, CU treatment caused the increase of the apoptotic cell population compared with the control treatment. Similarly, A549 cells exposed to CU17 showed increased populations of apoptotic cells when compared with the control treatment (Figure 7c,d). These results indicated that both CU and CU17 induced apoptosis of lung cancer cells in a dose-dependent manner.

To investigate the factors involved in apoptosis induction, the expression levels of pro-apoptotic (Bax), anti-apoptotic (Bcl-2) and ERK signaling (pERK 1/2) proteins were examined by Western blot analysis. As shown in Figure 8, both CU and CU17 treatments enhanced Bax expression in a dose-dependent manner. The level of Bcl-2 was greatly reduced in CU and CU17 treatments. Furthermore, p-ERK1/2 levels were dramatically decreased, suggesting that the suppression of the ERK1/2 signaling pathway may be involved in the induction of apoptosis by CU and CU17 in lung cancer cells.

### 2.5. CU17 Enhances the Antiproliferative Effect of Gemcitabine (Gem) on Human Lung Cancer A549 Cells

As mentioned previously, both CU and CU17 could inhibit the proliferation of A549 cells. Therefore, we investigated whether CU or CU17 could synergize with Gem to inhibit the proliferation of A549 cells. The combined effect of Gem and CU or CU17 on cell viability of human lung cancer A549 cells was investigated. The inhibition rate of fixed dose Gem (IC_20_ = 0.68 and 0.35 µM, IC_30_ = 1.30 and 0.52 µM at 48 and 72 h, respectively) was combined with varying concentrations of CU or CU17 for 48 and 72 h. As shown in Figure 9, the co-treatment between Gem and CU17 more greatly reduced the survival of A549 cells than either treatment alone. 

Additionally, A549 cells were exposed to CU17 and Gem for 48 and 72 h, and their survival was reduced by approximately 42 to 50%. On the other hand, CU17 treatment alone suppressed A549 cell proliferation by approximately 16%, whereas Gem treatment alone at sub-toxic doses (IC_20_ and IC_30_ values) reduced A549 cell viability by approximately 36 to 49% after exposures at 48 and 72 h, respectively. Moreover, the IC_50_ values of the combination treatments between Gem (IC_20_) and CU17 were 0.45 ± 0.13 and 3.24 ± 0.32 µg/mL, while the co-treatment of Gem (IC_30_) and CU17 had IC_50_ values of 0.37 ± 0.21 and 2.04 ± 0.05 µg/mL after being exposed for 48 and 72 h, respectively. Furthermore, CU single treatment inhibited cell viability by approximately 56 to 64%, whereas the co-treatment with Gem and CU inhibited cell survival by approximately 50% less than CU alone treatment. The IC_50_ values of combination treatment of Gem (IC_20_) and CU were 3.97 ± 0.40 and 3.30 ± 0.07 µg/mL, while the co-treatment of Gem (IC_30_) and CU had IC_50_ values of 3.48 ± 0.40 and 2.87 ± 1.00 µg/mL at exposure times of 48 and 72 h, respectively. These results indicated that CU17 could drastically enhance the chemotherapeutic effect of Gem on lung cancer cells more than the combination treatment of Gem and CU. 

### 2.6. Combination Index and Dose Reduction Index of Gem and CU or CU17 in Combination Treatments

The Chou–Talalay approach was used to estimate the combination index (CI) and dose reduction index (DRI) values to assess the type of drug interaction between Gem and CU or CU17. A concentration of Gem was fixed in co-treatments with different concentrations of CU or CU17 in A549 cells at sub-toxic concentrations (the dosages that cause growth inhibition approximately 20–30%). As shown in Table 2, the combination treatment of Gem and CU17 at 48 h exposure revealed a synergistic effect (CI = 0.13 ± 0.02 and 0.18 ± 0.00) in A549 cells. 

Additionally, these synergistic effects indicated a dose reduction of 7.91 to 15.94-fold for the Gem and 38.93 to 56.25-fold for CU17, whereas CU17 combined with Gem at 72 h exposure presented a nearly additive effect (CI = 1.18 ± 0.09 and 1.07 ± 0.07) in A549 cells. However, the combination treatment of Gem and CU exhibited CI values of 1.44 ± 0.14, 1.40 ± 0.15, 2.11 ± 0.40, and 2.20 ± 0.63, at 48 and 72 h exposures, respectively. The CI values of more than 1 indicated an antagonistic effect in A549 cells.

## 3. Discussion

Lung cancer is the leading cause of cancer-related mortality worldwide. Chemotherapy is the most common treatment option for lung cancer. However, chemotherapeutic drugs influence both malignant tumor cells and normal cells [1,2]. Therefore, searching for chemopreventive and antineoplastic agents that are both effective and less toxic is still necessary for therapeutic strategies. Curcumin (CU) is a polyphenol compound extracted from the rhizome of turmeric (*Curcuma longa*). CU is a nutraceutical exhibiting a wide range of biological actions that are beneficial to human health. Curcumin supplementation has been demonstrated to be effective against a variety of diseases and pathological conditions, including anxiety and depression disorders. Among the many potential therapeutic activities of CU are anticancer effects on various cancer cell lines [22,23]. Despite these benefits, the limitation of CU bioavailability is still a major issue. Some experts believe that further clinical studies on CU are unnecessary due to its unstable, reactive, and nonbioavailable characteristics [24]. The clinical application of CU has been severely restricted owing to its low bioavailability, instability, and rapid metabolism in vivo, all of which have been identified as major factors [25]. Hence, the investigation of novel chemically synthesized curcumin analogues with comparable biological properties has received much attention. CU has been extensively identified for its ability to inhibit numerous malignancies, contributing to cancer prevention [13]. However, the therapeutic effectiveness of CU is limited due to its chemical instability and rapid metabolism [16]. In the present study, we determined the anti-cancer activity of the curcumin derivative CU17 and explored its underlying antitumor mechanisms against human lung cancer cells. CU17 used in this study was successfully synthesized from CU according to previously described methods [21], and data demonstrating successful synthesis/purity check are provided in the Appendix A. The structure and function of histone proteins are altered by an imbalance between histone acetyltransferases (HATs) and histone deacetylases (HDACs), which affects cancer cell processes [26]. HDACs play an essential role in carcinogenesis and the high level of HDAC expression has been associated with lung cancer. Hence, HDACs have been identified as interesting targets in cancer therapy. HDAC inhibitors (HDACis) have been improved, and they currently constitute a highly promising treatment approach [27]. Here, we demonstrated that CU17 effectively inhibited HDAC activity in a dose dependent manner (Figure 2a) and the molecular docking test supported the in vitro results. In silico results revealed that CU17 could interact with numerous amino acids in the catalytic site of human HDAC class I and class II. Moreover, CU17 was also chelated to zinc ions, which function as cofactors for HDAC activity (Figure 3). Consistently, CU17 treatment caused hyperacetylation of histone H3 in A549 cells (Figure 2b). These results confirmed that CU17 has a stronger HDAC inhibitory effect in A549 cells, which supports its activity in vitro. Acetylation of histone could promote cell cycle arrest and prevent cell proliferation; therefore, inhibitors of histone deacetylases have been suggested as a possible epigenetic oncotherapy [28]. For instance, acetylation of H4 could trigger Raji cells to enter the G2/M cell cycle by sealing chromatin structures and suppressing the transcription of target genes such as CDK1 and cyclin B1 [29]. Quisinostat, as an HDAC inhibitor, has been shown to induce cell cycle arrest and apoptosis in A549 lung cancer cells by maintaining H3 and H4 acetylation [30]. Additionally, HDAC inhibitors caused an increase in the expression of death receptors and ligands involved in extrinsic apoptosis pathways [31]. Pro-survival proteins, such as Bcl-2 and Bcl-1, which preserve mitochondrial integrity, are downregulated by HDACis, while pro-apoptotic proteins, such as Bim, Bak, and Bax, which act as monitors of oxidative responses and activate the intrinsic pathway, are upregulated by HDACis [32,33,34]. Furthermore, hyperacetylation has also been demonstrated to maintain p53 stability in cancer cells, inducing cell-cycle arrest and activation of the pro-apoptotic genes [35].

In this study, both CU and CU17 inhibited the proliferation of A549 cells. However, CU exhibited a more cytotoxic effect against non-cancer Vero cells. The amino derivatives of CU, such as CU16, CU17, CU18 and CU19, were synthesized by chemical modification of CU [21]. They displayed a lower physiochemical instability but stronger pharmacological activities than the CU analog. Besides, they have HDAC inhibitory activity and antiproliferative activity in a variety of cancer cell lines (Hela, HCT116, MCF-7 and HCT116 cells) [21]. Furthermore, HHMM-41, a new CU derivative, inhibited lung cancer cell growth more effectively than CU with no cytotoxicity effect on normal cells [16]. JZ534 CU derivative had stronger anti-lung cancer efficacy than the CU analog, suppressing lung cancer cell proliferation and differentiation as well as triggering cellular apoptosis [17]. WZ35 demonstrated high chemical stability and antitumor activity in several cancer cell lines. WZ35 was able to cause G2/M phase arrest and cell death in gastric cancer cells by activating ROS-dependent ER stress and JNK mitochondrial pathways [36].

CU has been shown to have cytotoxicity against non-small-cell lung cancer (NSCLC) by inhibiting cell proliferation, inducing apoptosis and arresting cells in the G2/M phase [37]. The extracellular signal-regulated kinase (ERK) pathway is crucial for cancer cell chemosensitivity [38]. CU affects the ERK 1/2 pathway specifically, resulting in a 75% decrease in its expression [39]. In this study, CU17 inhibited the growth of lung cancer A549 cells through the same underlying mechanism as CU (Figure 5, Figure 6, Figure 7 and Figure 8). CU and CU17 caused G2/M phase arrest in A549 cells, as shown by increasing percentages of cells in the G2/M phase (Figure 5). Furthermore, both CU and CU17 inhibited the survival of A549 cells via apoptosis induction (Figure 7). Nevertheless, CU17 triggered apoptosis in A549 cells through suppressing p21 and p53 levels (Figure 6). Although p53-dependent DNA damage-induced apoptosis is well understood as a significant response to a variety of anti-cancer medications, p53-independent apoptosis has also been identified in numerous cancer cell lines. Furthermore, apoptosis has also been discovered to be caused by a DNA damage mechanism that is not dependent on p53. There may be backup systems for p53-independent DNA damage-induced apoptosis, such as p53 homologs p63 and p73, CD95L (FAS, TNF-, Fas ligand [FasL]/Apo1L/Trail/Apo2L, Apo3L) inducing apoptosis, or p53-independent apoptosis via Bcl-2 degradation [40]. Consistent with our results, the treatments with CU17 down-regulated Bcl-2 expression in A549 cells according to Western blotting. Moreover, CU17 up-regulated the expression of Bax protein (Figure 8). A previous report suggested that the increased Bax to Bcl-2 ratio could induce the mitochondrial membrane potential to rupture, resulting in the release of cytochrome c and cell apoptosis [41,42]. Hence, apoptosis induction in A549 cells was confirmed by the increasing Bax/Bcl2 ratio in the CU17 treatment. According to earlier reports, the phosphorylation of Bcl2 has been widely reported to halt cells at the G2/M phase of the cell cycle [43,44]. Our data showed that the expression of Bcl2 proteins decreased after treatment with the HDAC inhibitor CU17, resulting in cell cycle halting at the G2/M phase in A549 cells. 

Although p21 activation is mainly responsible for HDAC inhibitor-induced cell cycle arrest, HDAC inhibitors also involve a p21-independent mechanism in cell cycle arrest. Trichostatin A stimulates the p15Ink4b gene, an INK4 family protein, and causes cell cycle arrest in colon cancer cells deficient in p21 [45]. Trichostatin A also caused an increase of acetylated histones H3 and H4 in HepG2 and Huh-7 cells, as well as considerable growth suppression and G0/G1 phase arrest in hepatoma cells [46]. In this study, the positive control cisplatin, a DNA cross-linking agent, was shown to cause p53-dependent DNA damage-induced apoptosis and the cell cycle arrest pathway (Figure 5 and Figure 7). Western blot analysis showed that expression levels of cell cycle-related proteins, p21 and p53, were upregulated after cisplatin treatment (Figure 6). Moreover, cisplatin upregulated Bax expression but had no effect on the expression of Bcl-2 in A549 cells (Figure 8). The anticancer mechanism of cisplatin is widely considered to be that it causes DNA damage by forming DNA adducts [47]. The p53 protein controls the cellular response to DNA damage, causing the transcription of several p53 response genes involved in cell cycle regulation, DNA repair, and apoptosis to be activated. The upregulation of p53 expression is associated with induction of p21 proteins [48]. DeHaan et al. have reported that the level of Bax, Bcl-2, and Bcl-XL are not altered by cisplatin treatment [49]. Normally, Mdm2 binds to the p53 protein, which is then degraded through a ubiquitin-dependent proteolytic process [50]. Additionally, ERK phosphorylates p53 protein, leading to a reduction of Mdm2 and p53 binding. Interference with the binding of Mdm2 and p53 causes an increase in the accumulation of p53 protein [51,52]; hence, the cisplatin-induced ERK phosphorylation upregulated accumulation and function of p53 [41]. In this study, pERK1/2 level was downregulated after treatment with CU17 in A549 cells. The ERK/MAP-kinase pathway is engaged in a variety of signal transduction processes [53]. There is a correlation between Bcl-2 degradation and the level of pERK1/2. The oxidative stress response was also associated with TNF-activated Bcl-2 proteolytic degradation and a decrease of ERK/MAP-kinase activity [54].

Presently, chemotherapy is the major treatment option for lung cancer. However, chemotherapeutic drugs for lung cancer treatment still cause many side effects. Hence, the therapeutic efficacy of Gem should be improved further. Hatami et al. demonstrated that Gem had a significantly stronger therapeutic effect in A549 and H1299 cells when it was combined with GA [2]. Ginsenoside Rg3 in combination treatment with gemcitabine has been shown to reduce angiogenesis and lung cancer growth [55]. In this study, we evaluated whether CU or CU17 can potentiate Gem-induced anticancer efficacy. The results demonstrated that CU17 in combination with Gem exhibited a synergistic effect in lung cancer cells. In contrast, CU in combination with Gem showed an antagonistic effect in A549 cells (Figure 9). These results suggest that CU17 could be a candidate in combination with Gem for lung cancer treatment.

## 4. Materials and Methods

### 4.1. Materials

RPMI-1640 medium, trypsin-EDTA and penicillin/streptomycin were acquired from Thermo Fisher Scientific Inc. (Grand Island, NY, USA) while fetal bovine serum (FBS) was purchased from Cytiva (Kremplstrasse, Pasching, Austria). 3-(4,5-dimethylthiazol-2-yl)-2,5-diphenyltetrazolium bromide (MTT) was obtained from Invitrogen (Eugene, OR, USA) whereas Annexin V-FITC was from Biolegend (San Diego, CA, USA). Cisplatin, Gemcitabine hydrochloride (Gem) and Propidium iodide (PI) were acquired from Sigma-Aldrich Corporation (St. Louis, MO, USA). The primary antibodies against p53, Bcl-2, Bax, Ac-H3, p21, pERK1/2, ERK1/2 and the secondary antibodies including anti-mouse IgG and anti-rabbit IgG conjugated to horseradish peroxidase were purchased from Cell Signaling (Beverly, MA, USA). In addition, turmeric rhizome powder was derived from the herbal drugstore at a local market in Khon Kaen province, Thailand.

### 4.2. Cell Lines and Culture Condition

Human lung cancer (A549) cells were kindly provided by Assoc. Prof. Dr. Arunporn Ittharat (Department of Applied Thai Traditional Medicine, Faculty of Medicine, Thammasat University, Bangkok, Thailand). The non-cancer (Vero) cell line was kindly provided by Assoc. Prof. Dr. Sahapat Barusrux (Research of Excellence Center for Innovative Health Products (RECIHP), School of Allied Health Science, Walailak University, Nakhon Si Thammarat, Thailand). The cells were maintained in RPMI-1640 medium supplemented with 10% fetal bovine serum, penicillin (100 U/mL), and streptomycin (100 µg/mL) (Gibco-BRL, Gaithersburg, MD, USA). The cells were incubated at 37°C in a humidified atmosphere with 5% CO_2_.

### 4.3. The Extraction and Isolation of Curcumin

Curcumin was extracted and isolated from rhizome of turmeric by the method of Kumboonma [21]. Briefly, 600 g of turmeric powder was extracted in 1000 mL of dichloromethane (CH_2_Cl_2_) three times (1000 mL/time). Afterwards, the crude extract was concentrated under reduced pressure in a rotary evaporator to yield 90 g of dichloromethane extract. The dichloromethane extract was subjected to silica gel column chromatography. Then, hexane, ethyl acetate (EtOAc), and methanol (MeOH) were used to elute the extract under the increasing polarity of the selection system. The elution solvents were collected and separated into four fractions (DT1-DT4) by using TLC. DT1 was an oil fraction. The fraction DT2 was separated into three subfractions (DT2-1 to DT2-3) by using flash silica gel column chromatography and a gradient elution of hexane-EtOAc (10:0 to 5:5). DT2–1 and DT2–2 were purified using preparative thin layer chromatography with hexane-EtOAC (9:1) as the mobile phase, yielding demethoxycurcumin (0.05 g) and dihydrocurcumin (0.04 g) as a solid yellow color. DT2–3 was isolated with silica gel column chromatography and subsequently eluted with CH_2_Cl_2_-MeOH (9:1) to give curcumin (20 g) as a solid orange color. DT3 was exposed to silica gel column chromatography and then eluted by gradients CH_2_Cl_2_-MeOH solvent (10:0 to 7:3) to produce two subfractions (DT3–1 and DT3–2). DT3–2 was purified by using column chromatography with CH_2_Cl_2_-MeOH (9:1) as the elution solvent, affording bisdemethoxycurcumin (1.0 g). DT4 was separated into two subfractions (DT4–1 and DT4–2) by using a silica gel column and gradient elution of CH_2_Cl_2_-MeOH solvent (10:0 to 7:3). DT4–1 and DT4–2 were purified by using preparative thin layer chromatography with CH_2_Cl_2_-MeOH (9:1) as a mobile phase, yielding hydroxycurcumin (0.012 g). The NMR data of all products were similar to those reported by Venkateswarlu [56].

### 4.4. Curcumin Derivative CU17 Synthesis

Curcumin derivative CU17 was synthesized in our laboratory according to the method described by Kumboonma [21]. Briefly, curcumin (100 mg: 0.2715 mmol) was dissolved in ethanol (5 mL). Subsequently, curcumin solution was added with 2-aminothiophenol and stirred at room temperature. The combined solution was refluxed for 6 h. After completion of the reaction, the combined solution was filtered through Whatman no. 1 filter paper and was washed with ethanol (10 mL) and dried with anhydrous sodium sulfate. The solvent was removed by evaporation under reduced pressure and the crude product was purified by column chromatography and then eluted with 5% MeOH in CH_2_Cl_2_ to afford a yellow solid of CU17 (110 mg: 0.2228 mmol) (82% yield).

### 4.5. In Vitro HDAC Inhibitory Activity

Inhibition of HDAC activity in vitro was determined by using the Fluor-de-Lys HDAC Fluorometric Activity Assay Kit (Biomol, Enzo Life Sciences International, Inc., New York, NY, USA). Various concentrations of CU17 were evaluated for their HDAC inhibitory activity against the HDAC enzymes from the HeLa nuclear extract provided with the kit. The HeLa nuclear extract, buffer, and CU17 were added into a 96-well microplate and were incubated at 37 °C for 5 min. Afterwards, substrates were added to each well and were incubated at 37 °C for 15 min. HDAC reactions was stopped by the addition of developer. Subsequently, the plate was incubated at room temperature for 25 min. Samples were measured using a Spectra Max M5 fluorescence microplate reader (Molecular Devices) with excitation at 360 nm and emission at 460 nm. TSA was applied as the positive control. A decrease in fluorescence signal indicated an inhibition of HDAC activity. All experiments were carried out in triplicate.

### 4.6. Docking of CU17 to HDAC Structural Model

The interaction between the CU17 and HDAC was analyzed using AutoDock tools. The crystal structures of HDAC1, HDAC2, HDAC3, HDAC8, HDAC4, HDAC6 and HDAC7 [PDB entry code: 4BKX, 3MAX, 4A69, 1T64, 2VQW, 6UO2 and 3C0Z, respectively] were obtained from the Protein Data Bank (http://www.rcsb.org/pdb) (accessed on 1 March 2022) as well as used for molecular docking studies. All water and non-interacting ions and ligands were eliminated. Subsequently, all missing hydrogen and side-chain atoms were added using the ADT program. Gasteiger charges were calculated for the system. For ligand setup, the molecular modeling program Gaussview was used to build the ligands. The ligands were optimized with the AM1 level by using Gaussian 03W. A grid box size of 60 × 60 × 60 points with a spacing of 0.375 Å between the grid points was implemented and covered almost the entire HDAC protein surface.

### 4.7. Cell Viability Assay

The survival of cells was determined by MTT assay. A549 and Vero Cells were seeded into 96-well plates at a density of 8 × 10^3^ cells/well for 24 h. Subsequently, cells were exposed with various concentrations of CU or CU17 and solvent (0.5 % DMSO + 0.5% ethanol: solvent control) at 37 °C for 24, 48, and 72 h. After incubation of treatment, old medium was replaced with fresh medium containing MTT and was incubated at 37 °C for 2 h. Then, formazan was dissolved with DMSO and incubated at room temperature for 15 min. Finally, the absorbance was measured at a wavelength of 550 nm and a reference wavelength of 655 nm by a microplate reader (iMark™ Microplate Absorbance Reader, Bio-Rad, Hercules, CA, USA). The percentages of cell viability were calculated by following equation below.
(1)$$\mathrm{Cell}\;\mathrm{viability}\;(\%)=\frac{A_{550\mathrm{treatment}}-A_{650\mathrm{treatment}}}{A_{550\mathrm{solvent}\;\mathrm{control}}-A_{650\mathrm{solvent}\;\mathrm{control}}}\times 100.$$
where A is the absorbance.

### 4.8. Cell Cycle Progression

A549 cells were cultured in 60 × 15 mm^2^ dish plates at a density of 1 × 10^6^ cells/dish for 24 h. Subsequently, cells were exposed with three concentrations of CU or CU17 at 37 °C for 24 h. After treatment, cells were harvested, washed with 1X PBS containing 1% glucose and then fixed with 70% ethanol and incubated for 1 h on ice. Cells were washed with cold 1X PBS and incubated with RNase A (0.1 mg/mL) at 37 °C for 30 min. After incubation, cells were stained with PI in the dark and incubated at room temperature for 45 min. The stained cells were determined by a BD FACSCanto II flow cytometer. The percentages of different cells cycle phases were analyzed by the BD FACSDiva software.

### 4.9. Cellular Apoptosis Detection

Induction of apoptosis was determined by Annexin V and PI double staining assay. A549 cells were plated in a 60 × 15 mm^2^ dish at a density of 1 × 10^6^ cells/dish for 24 h. After that, cells were exposed as mentioned above in cell cycle progression. After treatment, cells were harvested, washed with cold 1X PBS, resuspended with 1X Annexin binding buffer, and then stained with Annexin V/PI. They were subsequently incubated in the dark at room temperature for 15 min. Cells were gently mixed with 1X Annexin binding buffer after staining and kept on ice. The apoptotic cells were quantified by BD FACSCanto II flow cytometer (Becton Dickinson, San Jose, CA, USA) and analyzed by the BD FACSDiva software supported by the Research Instrument Center (RIC), Khon Kaen University, Thailand. 

### 4.10. Western Blot Analysis

A549 cells were seeded at a density of 1 × 10^6^ cells/dish in 60 × 15 mm^2^ dish for 24 h. After that, cells were exposed as mentioned above in apoptosis analysis. Afterwards, cells were washed with cold 1X PBS and lysed with RIPA buffer containing a protease inhibitor cocktail (Amresco Inc., Solon, OH, USA), incubated on ice for 20 min. After that, cells were centrifuged at 12,000 rpm for 10 min at 4 °C and then, the supernatant was collected. The total protein extracted from cells was measured by Bio-Rad protein assay (Bio-Rad, Hercules, CA, USA). The equal amounts of cellular proteins (15–30 µg) were denatured in 1X loading dye, separated by using 12.5% sodium dodecyl sulphate-polyacrylamide gel electrophoresis (SDS-PAGE), and then transferred to a Polyvinylidene Fluoride (PVDF) membrane (Merck KGaA, Darmstadt, Germany). The blot was blocked with 3% skim milk in 1X PBST for 1 h and afterwards was exposed to various primary antibodies against p53 (#2524, diluted 1:1000), Bcl-2 (#4223, diluted 1:1000), Bax (#2772, diluted 1:1000), H3 (#9649, diluted 1:1000), p21 (#2946, diluted 1:1000), pERK1/2 (#9102, diluted 1:1000), ERK1/2 (#9107, diluted 1:2000) (Cell Signaling, Beverly, MA, USA) at 4 °C overnight. The next day, the blot was washed twice with cold 1X PBST for 2 min and then incubated with an anti-mouse (#7076, diluted 1:2000) or anti-rabbit (#7074, diluted 1:2000) secondary antibody at room temperature for 2 h. After incubation, the blot was washed twice with cold 1X PBST and PBS, respectively, each time for 2 min. Finally, the blot was visualized using enhanced chemiluminescence reagent (Bio-Rad, Hercules, CA, USA) and immunoreactive bands were examined using X-ray films. The intensity of relatives was quantified by the ImageJ program and total ERK1/2 was used as a loading control in the Western blot analysis.

### 4.11. Statistical Analysis

All the data from three separate experiments are shown as the mean ± standard deviation (SD) or standard error of the mean (SEM). Statistical analysis was performed using the statistical program IBM SPSS version 26.0 for Windows (SPSS Corporation, Chicago, IL, USA). One-way ANOVA analysis was employed to compare significant differences between the solvent control and treatment groups. Statistical significance was considered at *p* < 0.05.

## 5. Conclusions

CU17 inhibited HDAC activity both in vitro and in silico. Our results indicate that CU and CU17 can cause G2/M phase cell cycle arrest and p53-independent apoptosis. The antiproliferative effect of CU or CU17 treatment was increased by downregulating anti-apoptosis protein (Bcl-2) expression and upregulating pro-apoptosis protein (Bax) expression. The growth inhibition of A549 cells was also caused by an oncogenic cell signaling mechanism involving pERK1/2 inactivation. CU had a stronger inhibitory effect on lung cancer cells, while CU17 had a less cytotoxic effect on non-cancer cells. Therefore, CU17 is a great alternative for lung cancer treatment. Importantly, CU17 and gemcitabine combination treatment demonstrated synergistic anti-cancer effects in A549 cells. Hence, these results suggest that CU17 can enhance gemcitabine’s antiproliferative effect against lung cancer cells.

## Figures and Tables

**Figure 1 molecules-27-04014-f001:**
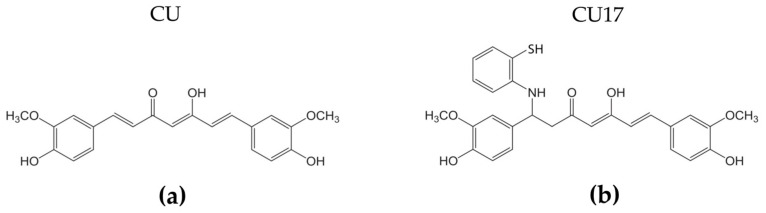
(**a**) Structure of curcumin (CU) and (**b**) curcumin derivative CU17.

**Figure 2 molecules-27-04014-f002:**
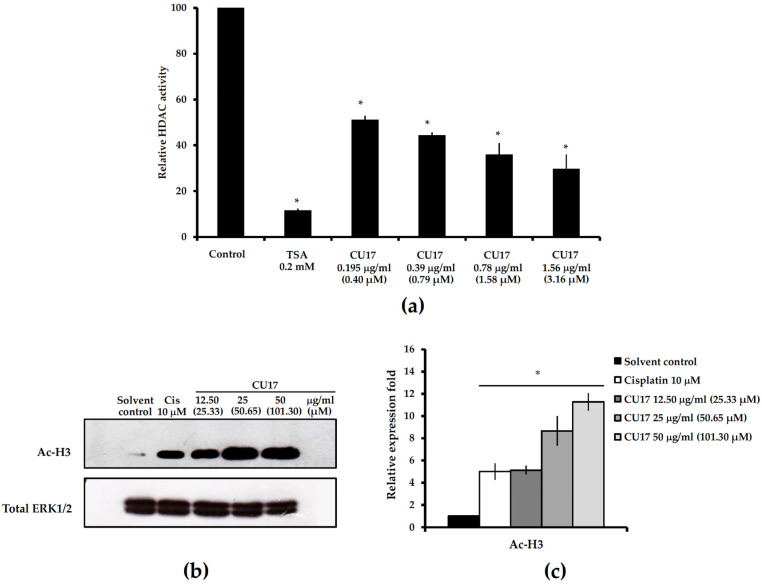
HDAC inhibitory activity of CU17. (**a**) In vitro HDAC inhibitory activity of CU17 was analyzed by the Fluor-de-Lys HDAC Fluorometric Activity Assay Kit. Bar graphs were expressed as relative HDAC activity with respect to the control (DMSO). Trichostatin A (TSA) was used as a positive control. Each value represents the mean ± SD from three independent experiments. (**b**) Western blot analysis of hyperacetylation of histone H3 in A549 cells after exposure to varying concentrations of CU17 for 24 h. DMSO: ethanol (1:1, *v*/*v*) and cisplatin (10 µM) were used as negative and positive controls, respectively. Total ERK1/2 was used as a loading control for Western blotting. (**c**) The relative fold of protein expression was calculated using the intensity of the protein band in comparison to a loading control and shown as a bar graph. Bar graph displayed the mean from three independent experiments. * *p* < 0.05 indicates a significant difference between the treatment and solvent control.

**Figure 3 molecules-27-04014-f003:**
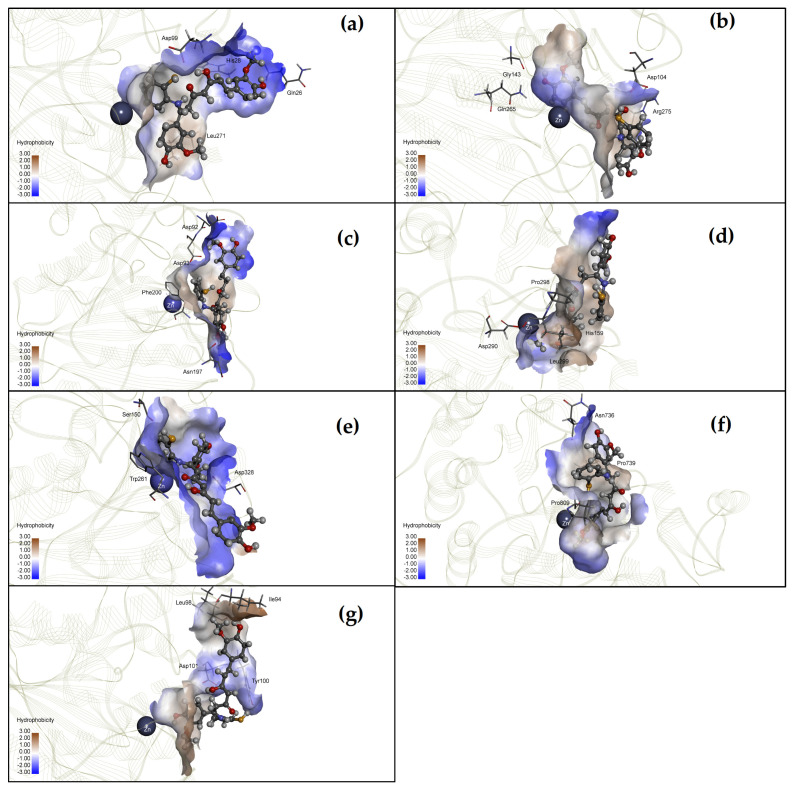
The interaction between CU17 and the active site of (**a**) HDAC1, (**b**) HDAC2, (**c**) HDAC3, (**d**) HDAC4, (**e**) HDAC6, (**f**) HDAC7, and (**g**) HDAC8.

**Figure 4 molecules-27-04014-f004:**
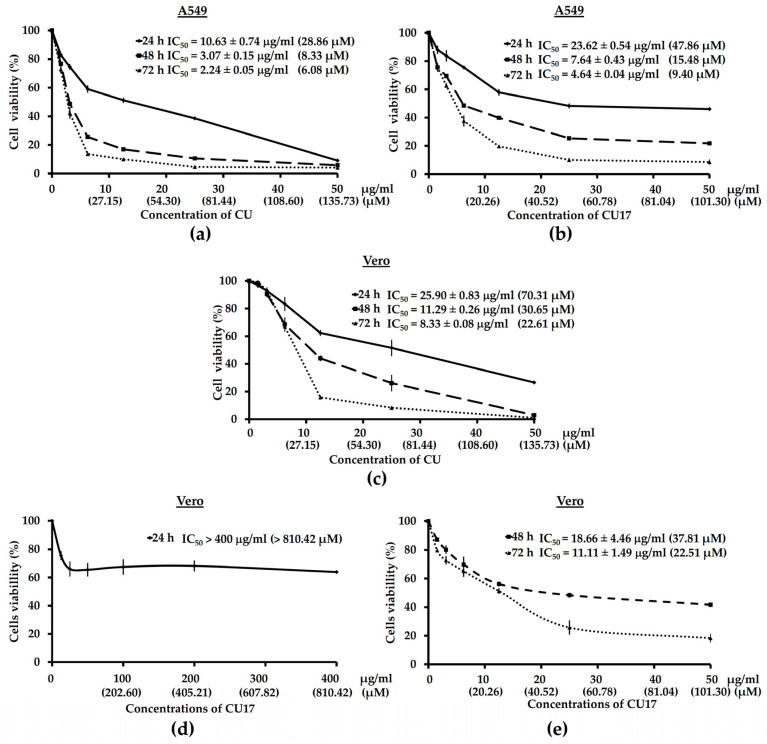
Effect of CU and CU17 on the proliferation of A549 cells (**a**,**b**) and non-cancer Vero cells (**c**–**e**) treated for 24, 48, and 72 h. Antiproliferative activity was determined by MTT assay. Data are shown as the percentage of cell viability compared with the solvent control (0 µg/mL), which was defined as 100%. The IC_50_ values are presented as the mean ± SEM from three independent experiments.

**Figure 5 molecules-27-04014-f005:**
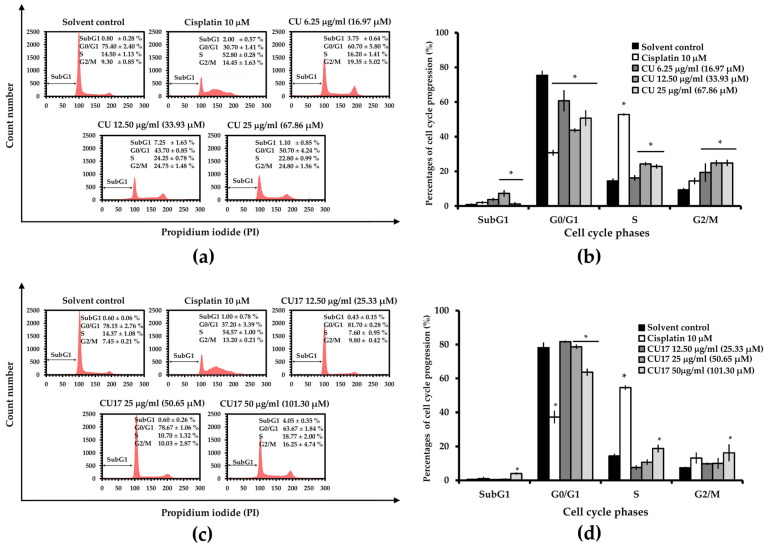
Effect of CU and CU17 on cell cycle progression of A549 cells for 24 h. (**a**,**c**) Cellular DNA histograms exhibit the cell cycle distribution of A549 cells after CU or CU17 treatment. (**b**,**d**) Bar graphs displayed the mean of percentages of cell cycle distribution from three independent experiments. A549 cells treated with DMSO: ethanol (1:1, *v*/*v*) and cisplatin (10 µM) were used as negative and positive controls, respectively. * *p* < 0.05 indicates a significant difference between the treatment and solvent control.

**Figure 6 molecules-27-04014-f006:**
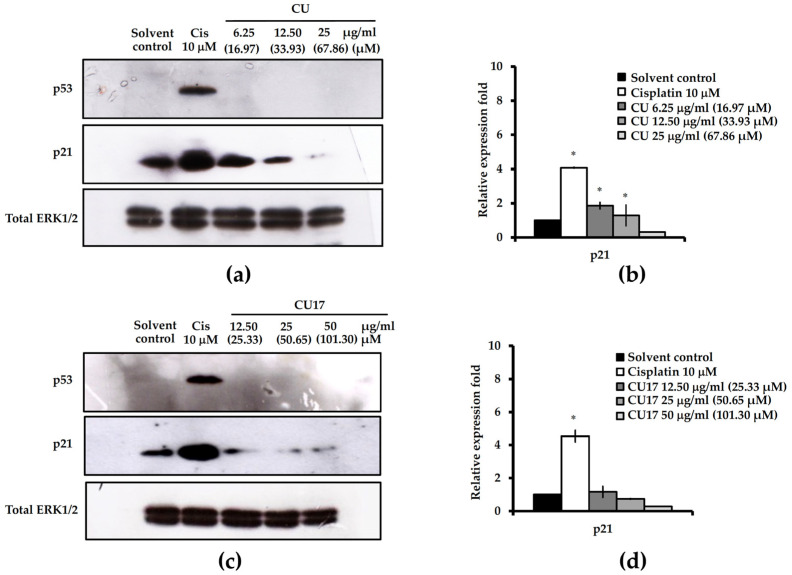
Western blot analysis of cell cycle-related proteins in A549 cells for 24 h exposure. (**a**,**c**) Expressions of cell cycle-associated proteins (p53 and p21) treated with CU or CU17 were determined, and the cells were exposed to DMSO: ethanol (1:1, *v*/*v*) and cisplatin (10 µM) as negative and positive controls, respectively. Total ERK1/2 was used as a loading control for Western blotting. The representative blots are from one experiment. (**b**,**d**) The relative fold of protein expression was calculated using the intensity of the protein band in comparison to that of a loading control and shown as a bar graph. Bar graph displayed the mean from three independent experiments. * *p* < 0.05 indicates a significant difference between the treatment and the solvent control.

**Figure 7 molecules-27-04014-f007:**
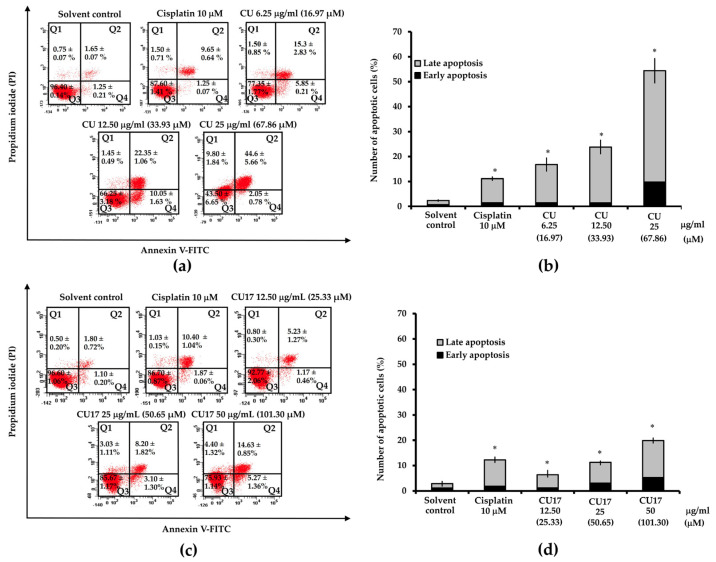
Effect of CU and CU17 on apoptosis induction in A549 cells for 24 h exposure. (**a**,**c**) The dot plots present the analysis of apoptosis induction on A549 cells after CU or CU17 treatment. (**b**,**d**) Bar graphs of the mean from three independent experiments displayed the percentages of apoptotic cells. A549 cells were treated with DMSO: ethanol (1:1, *v*/*v*) and cisplatin (10 µM) as negative and positive controls, respectively. * *p* < 0.05 indicates a significant difference between the treatment and solvent control.

**Figure 8 molecules-27-04014-f008:**
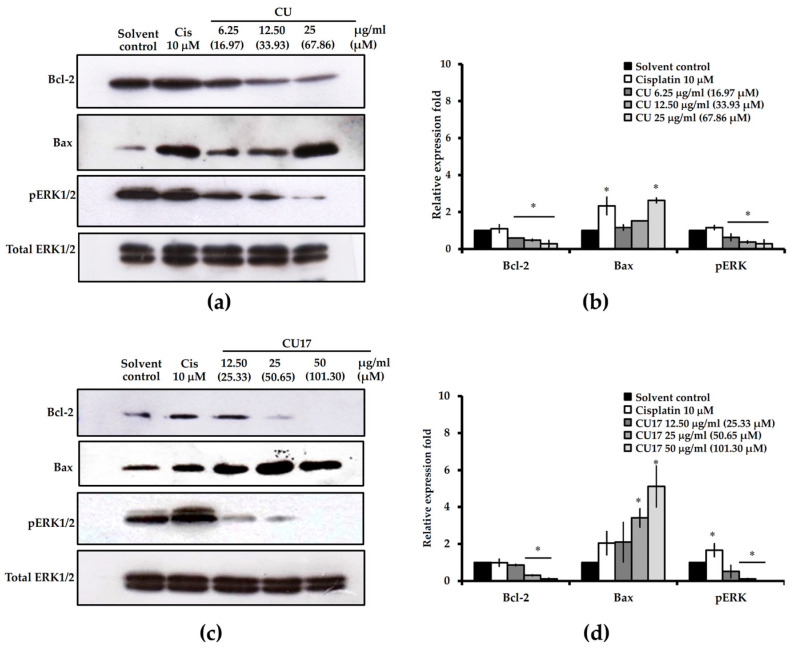
Western blot analysis of apoptosis-related and ERK signaling proteins in A549 cells. (**a**,**c**) Cells were exposed to various concentrations of CU or CU17 for 24 h. DMSO: ethanol (1:1, *v*/*v*) and cisplatin (10 µM) were used as negative and positive controls, respectively. Total ERK1/2 was used as a loading control for Western blotting. The representative blots are from one experiment. (**b**,**d**) The relative fold of protein expression was calculated using the intensity of the protein band in comparison to that of a loading control and shown as a bar graph. Bar graphs display the mean from three independent experiments. * *p* < 0.05 indicates a significant difference between the treatment and solvent control.

**Figure 9 molecules-27-04014-f009:**
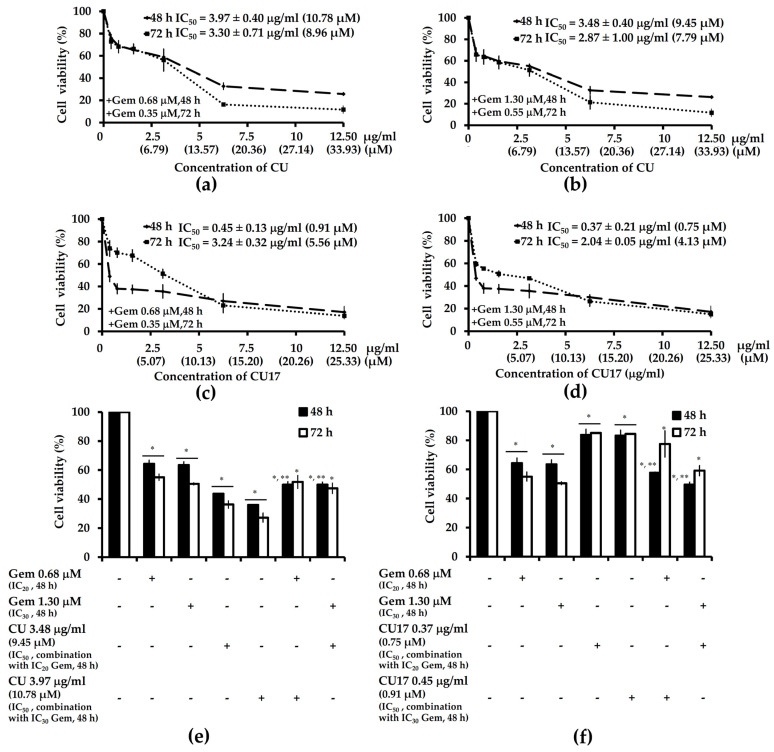
The effect of Gem and CU or CU17 combination treatments on proliferation of A549 cells. (**a**,**b**) Cells were treated with CU (0.39–12.5 µg/mL final concentration) and Gem (IC_20_ = 0.68 and 0.35 µM and IC_30_ = 1.29 and 0.52 µM final concentration) at 48 and 72 h, respectively. (**c**,**d**) Cells were treated with CU17 (0.39–12.5 µg/mL final concentration) and Gem (IC_20_ = 0.68 and 0.35 µM and IC_30_ = 1.29 and 0.52 µM final concentration) at 48 and 72 h, respectively. Data are shown as the percentages of the solvent control, which was defined as 100%. The IC_50_ values are expressed as the mean ± SEM from three independent experiments. (**e**,**f**) Bar graphs show percentages of cell viability. DMSO : ethanol (1:1, *v*/*v*) was used as a negative control. * *p* < 0.05 indicates a significant difference between the treatment and solvent control, while ** *p* < 0.05 indicates a significant difference between the alone and combination treatments.

**Table 1 molecules-27-04014-t001:** In silico histone deacetylase inhibitory activity of CU and CU17.

Compound	Class I							Class II						
	HDAC1		HDAC2		HDAC3		HDAC8		HDAC4		HDAC6		HDAC7	
	∆G	K*_i_*	∆G	K*_i_*	∆G	K*_i_*	∆G	K*_i_*	∆G	K*_i_*	∆G	K*_i_*	∆G	K*_i_*
**CU**	−6.96	7.88	−8.04	1.27	−9.82	0.06	−7.22	5.13	−9.11	0.21	−7.82	1.85	−8.12	1.11
**CU17**	−8.91	0.29	−7.64	2.51	−8.19	0.99	−7.96	1.45	−9.15	0.20	−9.21	0.18	−8.40	0.70

(∆G = kcal/mol, K*_i_* = µM).

**Table 2 molecules-27-04014-t002:** Combination index (CI) and dose reduction index (DRI) of the combination treatments between gemcitabine (Gem) and CU or CU17 against A549 cells.

Drug Combination	Exposure Time	A549 Cells			
		CI	DRI		
			Gem	CU	CU17
Gem 0.68 μM (IC_20_) + CU	48 h	1.44 ± 0.14	15.94	0.77	-
Gem 0.35 μM (IC_20_) + CU	72 h	2.11 ± 0.40	4.09	0.68	-
Gem 1.30 μM (IC_30_) + CU	48 h	1.40 ± 0.15	7.91	0.88	-
Gem 0.55 μM (IC_30_) + CU	72 h	2.20 ± 0.63	2.49	0.78	-
Gem 0.68 μM (IC_20_) + CU17	48 h	0.13 ± 0.02	15.94	-	56.25
Gem 0.35 μM (IC_20_) + CU17	72 h	1.18 ± 0.09	4.09	-	3.64
Gem 1.30 μM (IC_30_) + CU17	48 h	0.18 ± 0.00	7.91	-	38.93
Gem 0.55 μM (IC_30_) + CU17	72 h	1.07 ± 0.07	2.49	-	4.03

## Data Availability

The datasets created and/or analyzed during this investigation are accessible upon reasonable request from the corresponding author.

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
