# Peer review of "HDAC Inhibitory and Anti-Cancer Activities of Curcumin and Curcumin Derivative CU17 against Human Lung Cancer A549 Cells"

_molecules, 2022, doi:10.3390/molecules27134014_

Round 1
Reviewer 1 Report
- Add cancer statistics in the introduction section
- Add a few reference related to the importance of curcumin analogues in the last paragraph of introduction (e.g. https://doi.org/10.3390/plants10081559; https://doi.org/10.1016/j.arabjc.2015.04.011; https://doi.org/10.1016/j.actatropica.2018.12.033; https://doi.org/10.1080/17460441.2019.1614560; https://doi.org/10.1016/j.ejmech.2019.111631; https://doi.org/10.1016/j.phrs.2021.105489)
- Explain the synthetic protocol of the CU-17 in the results and discussion section
- Why the HDAC inhibitory activity of CU17 was not compared with the standard drug like SAHA etc.?
- Over all the manuscript is well written and clear to the reader.
Reviewer 2 Report
The manuscript entitled "HDAC inhibitory and anticancer activities of curcumin and curcumin derivative CU17 against human lung cancer A549 cells" documents the detail of the anti-cancer efficacy of CU17 and its potential for inhibiting HDAC. The manuscript is interesting however, there are a few concerns that need to be addressed. Please find my comments below.
- The introduction is too long and should be shortened with only vital information to form the background of the study.
- There are several grammatical errors throughout, and the manuscript requires extensive English editing. For example, in line #44, "action mechanism" should be "mechanism of action." Line #45 - Line #47 is confusing, and I recommend rephrasing it.
- Curcumin has been extensively studied for its various biological activities, including anti-cancer activity and HDAC inhibitory potential (in the A549 cell line). Additionally, reference #18 already documents the synthesis of CU17 and other derivatives. So I struggled to understand the novelty of this study. Hence, I recommend updating the introduction and discussion.
- Authors have synthesized CU17 according to previously described methods. So data demonstrating successful synthesis/purity check should be included in the supplementary material.
- Some of the western blots presented are overexposed and saturated. It is even more challenging to differentiate between two bands in some cases. For instance, Figure 8 (c) pERK1/2. There is a huge scope for improvement here. Also, in the supplementary file, it should be repeated 1,2,3 instead of 1,2,2
- Why have authors used ERK1/2 as the internal loading control over the most commonly used such as Beta-Acting or GAPDH? Is there a specific reason?
- I suggest adding the docking profile of CU as well in the in silico study.
Reviewer 3 Report
HDACs are promising cancer targets, and four U.S FDA-approved HDAC inhibitors have been used for cancer treatment. Like cis-platin in chemotherapy, these HDAC inhibitors also cause the severe side effects. Thanaset et al pursued curcumin and its derivative CU17 as the alternative HDAC inhibitors for treating cancers. The antiproliferative activity of these two compounds along with several other CU derivatives against multiple cancer cells have been demonstrated in their previous publication (Medicinal Chemistry Research 2019, 28, 1773-1782). The authors’ current work in illustrating the antiproliferative mechanism of CU and CU17 in comparison with cis-platin is interesting, however there are four major issues need to be addressed for reconsidering its publication in Molecules.
Major points
#1. The introduction needs to cover the significance of selectively inhibiting HDAC isoenzymes, which has been an important interest to address the side-effect issues of HDACis among HDACi community.
#2. CU is known as the pan-HDAC inhibitor. What about the potential of CU17 as the selective inhibitor against HDAC isoenzymes?
#3. Docking programs have serious flaws in predicting the binding constants as exemplified by the two reviews (J. Med. Chem. 2006, 49, 5912-5931; Molecules. 2018, 23, 1899).
#4. It is not surprising that CU17 causes the less damage to non-cancer cells as it has the less antiproliferative activity against cancer cells. This could also be interpretated as CU17 lacks drug efficacy on the contrary to the authors’ claim that CU could be the promising lung cancer treatment. Thus, it is necessary to reveal or underline the drug efficacy mechanism of CU17 that distinguishes itself from any other mediocre anticancer candidates, which have the weak antiproliferative activity and the less cytotoxicity to non-cancer cells but could not be developed as drugs.
Minor points
#1. Figure 3 needs to be improved for clearly showing which atom of CU17 is involved in the coordination with the catalytic zinc of HDACs and their coordination distance.
Round 2
Reviewer 1 Report
Manuscript is ok now
Author Response
Thank you very much for your kind review.
Reviewer 2 Report
The authors have made significant revisions, and the manuscript is in better shape.
Author Response

(The authors gave the same response as above.)

Reviewer 3 Report
The authors have improved the manuscript significantly, and the current form is ready for publishing in Molecule.
Author Response

(The authors gave the same response as above.)
